# Development and Analysis of Coding and Tailored Metamaterial for Terahertz Frequency Applications

**DOI:** 10.3390/ma15082777

**Published:** 2022-04-10

**Authors:** Tayaallen Ramachandran, Mohammad Rashed Iqbal Faruque, Mohammad Tariqul Islam, Mayeen Uddin Khandaker, Amal Alqahtani, D. A. Bradley

**Affiliations:** 1Space Science Centre (ANGKASA), Institute of Climate Change (IPI), Universiti Kebangsaan Malaysia, Bangi 43600, Malaysia; tayachandran@gmail.com; 2Department of Electrical, Electronic & Systems Engineering, Universiti Kebangsaan Malaysia, Bangi 43600, Malaysia; tariqul@ukm.edu.my; 3Centre for Applied Physics and Radiation Technologies, School of Engineering and Technology, Sunway University, Bandar Sunway 47500, Malaysia; mayeenk@sunway.edu.my (M.U.K.); d.a.bradley@surrey.ac.uk (D.A.B.); 4Department of Basic Sciences, Deanship of Preparatory Year and Supporting Studies, Imam Abdulrahman Bin Faisal University, Dammam 34212, Saudi Arabia; amalqahtani@iau.edu.sa; 5Centre for Nuclear and Radiation Physics, Department of Physics, University of Surrey, Guildford GU2 7XH, UK

**Keywords:** CST, coding metamaterial, radar-cross section, silicon, tailored metamaterial, terahertz frequency

## Abstract

This study represents the development and analysis of the types of metamaterial structures for terahertz frequency. Recently, investigations about unique coding metamaterial have become well-known among the scientific community since it can manipulate electromagnetic (EM) waves by utilizing various coding sequences. Therefore, several coding and tailored metamaterial designs were compared and numerically analyzed the performances in this research work. The 1-bit coding metamaterial made up of only “0” and “1” elements by adopting two types of unit cells with 0 and π phase responses were analyzed for the coding metamaterial. Moreover, for the numerical simulation analyses, the well-known Computer Simulation Technology (CST) Microwave Studio software was adopted. This investigation focused on the frequency ranges from 0 to 5 THz. On the other hand, the proposed designs were simulated to find their scattering parameter behavior. The comparison of coding and tailored metamaterial revealed slight differences in the RCS values. The coding metamaterial designs manifested RCS values less than −50 dBm^2^, while tailored metamaterial designs exhibited less than −60 dBm^2^. Furthermore, the proposed designs displayed various transmission coefficient result curves for both types of metamaterial. Moreover, the bistatic far-field scattering patterns of both metamaterial designs were presented in this work. In a nutshell, the 1-bit coding metamaterial with a unique sequence can influence the EM waves and realize different functionalities.

## 1. Introduction

Unconventional and synthetic material known as metamaterial possesses unique electromagnetic properties that are not found in natural properties. Many research works adopting metamaterial designs have been performed in recent decades, such as SAR reduction, sensors, satellite applications, metasurfaces, etc. [1,2,3,4,5,6,7]. Although metamaterial research has a long tradition, there have been limited coding metamaterial studies for terahertz frequencies conducted in recent years. There has been an increased recognition that more attention must be paid to this area of the field since coding metamaterial can manipulate EM waves by adopting various coding sequences with different bits. For example, in 1-bit coding sequences, the design comprises two types of unit cells to mimic “0” and “1” elements. On the other hand, 2-bit coding metamaterial typically possesses four types of unit cells to mimic “00”, “01”, “10”, and “11” elements. Therefore, for the following number of bits, the elements will change accordingly.

In 2014, Cui et al. [8] proposed digital metamaterials through two steps, including the construction of coding metamaterial by adopting 1-bit, and introducing unique metamaterial particles that have either a “0” or “1” response controlled by a biased diode. Moreover, 2-bit coding metamaterial has also been analyzed in this work. Moreover, Zhang et al. [9] reviewed the recent progress about the coding, and digital and programmable metamaterials in 2017. In this work, clear discussions on their capacities in manipulating real times EM waves and designing multi-functional devices were presented. In the same year, Wu et al. [10] proposed frequency coding metamaterial to control EM energy radiations with a fixed spatial coding pattern when changes in frequency happened. The study not only considered the different phase responses of the unit cells but also required various phase sensitivities. On the other hand, Cuong et al. [11] designed, simulated, and measured the broadband coding metamaterial for microwave absorber applications. The full-wave finite integration simulation in this work utilized a full-sized configuration instead of traditional unit cell boundary conditions. Moreover, four different coding metamaterial blocks such as 2 × 2, 3 × 3, 4 × 4, and 6 × 6 were analyzed at the initial stage. Following these optimization processes, they were used as building blocks to construct various 12 × 12 topologies with a realistic size scale.

In addition, studies of the metamaterial for terahertz frequencies are also becoming more well-known among researchers. In 2010, Iwaszczuk et al. [12] investigated angle- and frequency-resolved RCS measurements on an object that works in terahertz frequencies. The measurements were performed on a 5–10 cm sized aircraft model in polar and azimuthal configurations. Moreover, the measurements closely corresponded with conventional radar on full-sized objects. Moreover, Liu et al. [13] proposed a polarization-controlled anisotropic coding metamaterial for terahertz frequencies applications. In this work, an experimental analysis of an ultrathin and flexible metasurface was introduced by adopting specially designed coding elements. In 2015, Lui et al. [14] explored the feasibility of RCS measurement using a quantum cascade laser, which works in terahertz frequencies via laser feedback interferometry. The experiment data were compared with the numerical simulation that could be retrieved through the numerical fitting of the well-known excess phase equation. The RCS calculation and simulation of the three-dimensional conductive model based on the Runge–Kutta Exponential Time Differencing-Finite Difference Time Domain method was performed by Liu et al. [15] in 2021. Furthermore, the interaction of the terahertz frequency wave and magnetized plasma sheath was also addressed.

On the other hand, conventional metamaterial designs have also been utilized in RCS reduction values. Wen et al. [16] investigated RCS reduction analysis by proposing a metamaterial absorber structure that works in the microwave frequency band. The work was also validated by measuring the conducting plane and dihedral angle reflectors coated with both metamaterial and FR-4 layers. In addition, Zhang et al. [17] investigated the RCS reduction of the patch antenna by adopting a metamaterial absorber. The comparison of patch antenna patterns and reflection coefficients for loading and unloading the metamaterial was presented. In 2019, Fan et al. [18] reviewed the recent developments of the metamaterial and metasurfaces for RCS reduction applications. This review paper discussed the basic theory, working principles, design formula, and also experimental verification. Absorber properties by adopting metamaterial design in solar arrays with simultaneous high optical transparency and broadband microwave absorption were proposed by Kong et al. [19] in 2020. The effects of tailoring the reflection response of the metamaterial and employment of transparent substrate material such as indium tin oxide film and anti-reflection glass were discussed in this paper. All the previously published papers in this literature review indicate a clear concept of metamaterial in various application fields. However, RCS reduction application by adopting coding metamaterial design in terahertz frequencies is relatively new and can offer unique properties by conducting extensive research work. Therefore, in this work, several coding sequences based on the 1-bit coding patterns were analyzed, and a few tailored metamaterial structures were introduced which possess small traces of the coding metamaterial concept. Moreover, the 1-bit coding metamaterial consists of two types of unit cells to mimic the ‘0’ and ‘1’ elements. Therefore, the element “0” was constructed with the only substrate material. The square-shaped metamaterial structure was adapted with an optimized design for the element “1”. A total number of eight square rings and one square patch at the center were constructed on the silicon substrate material. Additionally, three types of coding sequences were proposed, and the radar cross-section (RCS) reduction values of the designs were numerically calculated. Moreover, four types of tailored metamaterials were also investigated to compare the results with coding metamaterials.

## 2. Coding Metamaterial Construction

The 1-bit coding metamaterial was utilized in this work where two types of unit cells were introduced to mimic the ‘0’ and ‘1’ elements with a 0 and π phase response, respectively. The selection process of these two elements was successfully performed by adopting the trial and error method. This simple method is capable of finding the best and all structures when a testable finite number of possible designs exists. Therefore, this work was initially designed with various square-shaped metamaterial structures for the proposed applications. On the other hand, the square structure was adopted since it is capable of exhibiting lower resonance responses for a specific number of rings and gaps between them. Hence, the number of rings, arrangement of rings, and the thickness of each ring were thoroughly analyzed by the trial and error method to find suitable unit cells. Consequently, at the end of this analysis, two unit cells designs were chosen, and the dimension and design details are explained in this section. Figure 1a,b illustrate the full dimension of the unit cell structure of the element “1” metamaterial. Moreover, the “0” element in this paper was adopted as only a substrate material known as silicon. This substrate material has a thickness of 15 µm and possesses a dielectric constant of 11.9 and tangent loss of 0.00025. The element “1” metamaterial design also has a similar type of substrate material. A square split ring resonator metamaterial design structure was adopted to mimic the element “1”. The thickness of this metamaterial design was adopted as 0.2 µm, and a gold material was used to construct them, which possesses conductivity (σ) of 4.56 × 107 S/m. In this design, a total number of eight square rings and one square patch at the center were constructed with distinct widths. Firstly, a 3.0 µm (t1) square ring was constructed on the substrate material, as shown in Figure 1a. The rest of the square rings were constructed with the gap between them of 1.5, 1.4, 1.0, 1.5, 0.5, 2.0, 0.2, and 0.3 µm, respectively, by utilizing the following ring widths from t2 to t9 (as shown in Table 1). Once the unit cell metamaterial design was completed, the sequences for coding metamaterial were analyzed. Several coding sequences were adopted in this work to gain the optimized RCS reduction values. Figure 1c,d illustrates the phase responses and phase difference of elements ‘0’ and ‘1’. The element ‘1’ has a phase response of more than 180° at several frequency ranges such as from 0.51 to 0.59 THz, 1.49 to 1.61 THz, 1.69 to 1.77 THz, 1.97 to 2.00 THz, and 4.20 to 4.36 THz. Therefore, the monostatic RCS calculation for the proposed coding metamaterial is suitably applied for terahertz applications. Further analysis details are discussed in Section 3.2.

All the simulation analyses in this research work were completed by utilizing Computer Simulation Technology (CST) Studio Suite 2019 software. The CST software was adopted because this program generally gives quick and precise outcomes for the virtual prototype before fabricating it to a real object. This will act as a physical trial and has shorter development cycles. These analyses are divided into two sections, namely scattering parameters and radar-cross section (RCS) reduction simulations. Both the sections utilized hexahedral mesh and time-domain solver. The unit cell and coding metamaterial design structure were placed in the middle of two waveguide ports to measure the reflection (*S*_11_) and transmission coefficient (*S*_21_) results. The position of these ports was adopted along the z-axis, which indicates a transverse electromagnetic mode. Furthermore, the y-axis was set as a perfect magnetic conductor, while the x-axis was set as a perfect electric conductor. This terahertz metamaterial research work mainly focused on the frequency ranges from 0 to 5 THz. This frequency range was set before the construction of the unit cell metamaterial design. By adopting the trial and error method, the construction of the design was modified accordingly to gain response in the selected frequency range. At the end of the first simulation section, the scattering parameters of the proposed design were produced. Following this discovery, the *S*_11_ and *S*_21_ results were used to calculate the effective medium parameters of the element “1” metamaterial design. Therefore, MathWorks MATLAB R2021 software was utilized to calculate these parameters such as permittivity (*ε*), permeability (*μ*), refractive index (*n*), and impedance (*z*) values by adopting the Robust method [20,21,22]. This software is an advanced and powerful numerical computing application that offers strong high-level scripting language for scientific and mathematical problems. Equations (1)–(5) described below indicate the retrieval equations of *z*, *n*, *ε*, and *μ*, respectively. On the other hand, High-Frequency Structure Simulator (HFSS) software was adopted to validate the CST numerical simulation result of the unit cell metamaterial design. Therefore, the corresponding methods in CST numerical simulation were adopted for this software.
(1)$$z=\pm \sqrt{\frac{{\left(1+S_{11}\right)}^{2}-S_{21}^{2}}{{\left(1-S_{11}\right)}^{2}-S_{21}^{2}}}$$
(2)$$n=\frac{1}{k_{0}d}\{[{\left[ln\left(e^{ink_{0}d}\right)\right]}^{\prime \prime }+2m\pi ]-i{\left[ln\left(e^{ink_{0}d}\right)\right]}^{\prime }\}$$
(3)$$e^{ink_{0}d}=\frac{S_{21}}{1-S_{11}\frac{z-1}{z+1}}$$
(4)$$\varepsilon =\frac{n}{z}$$
(5)μ=nz

The symbols such as ‘ and ‘’ in Equation 2 denote the real part and imaginary part operators, respectively. The next section is the calculation of RCS reduction values for the proposed coding metamaterial designs. All the RCS simulations were performed in CST software by adopting hexahedral mesh. The monostatic RCS for the parameterized plane wave was identified by adopted under the far-field results in CST software. Moreover, the RCS reduction values were displayed in the dBm^2^ unit for all coding metamaterial designs.

## 3. Results and Discussion

This section is divided into three main sections which analyze the performance of element “1”, coding metamaterial, tailored metamaterial designs, and finally the scattering pattern comparison of all proposed metamaterial structures. Both the coding and tailored metamaterial designs were explored by comparing the scattering parameters and RCS reduction values. In each section, three different coding sequences were adopted. Moreover, these analyses also studied the performance chances by adding a multi-layered constraint variable that offers unique changes in outcomes.

### 3.1. Element “1” Metamaterial

Figure 2 illustrate the physical properties of element “1” metamaterial design such as reflection and transmission coefficients, permittivity, permeability, and refractive index. The unit cell metamaterial was simulated by utilizing CST and HFSS software to validate the outcomes (as illustrated in Figure 2a). The comparison demonstrated minor discrepancies between both methods, and therefore it can be accepted for practical applications. Overall, the simulation in CST software exhibited seven resonance frequencies in the range from 0 to 5 THz which had more than −15 dB magnitude values. For example, 0.72, 2.06, 2.43, 2.95, 3.15, 3.36, and 4.31 THz resonance frequencies had magnitude values of −26.81, −21.95, −16.25, −16.06, −20.71, −22.88, and −15.81 dB, respectively. On the other hand, the permittivity and refractive index values, as illustrated in Figure 2b,d, revealed similar responses within the frequency range from 0 to 1.5 THz. Moreover, most of the responses of permeability values occurred in the range between 3.5 and 5 THz and reached a maximum magnitude value of more than −30 dB.

### 3.2. Coding Metamaterial Sequence

A bit is generally known as the smallest unit of data in a computer and has a single binary value such as “0” or “1”. Therefore, in this work, the 1-bit was adopted to construct various coding metamaterial designs. However, the number of lattices was used to decide the properties of the constructed coding metamaterial structure. The main aim of this research work was to investigate coding metamaterial on a smaller scale, also known as the miniaturization concept, to realize optimized RCS reduction values. Therefore, six lattices were adopted in this research work, and all the optimized codes for the analysis were based on 6 × 6 with “0” or “1” elements. Three types of coding sequence patterns were analyzed in this section, as illustrated in Table 2. Preliminary results revealed that the optimization of the coding sequence by utilizing “0” and “1” elements can gain various RCS reduction patterns.

#### 3.2.1. Transmission Coefficient

Figure 3a–c demonstrate the *S*_21_ results of the three different coding sequences with their respective substrate layer comparisons. On the other hand, the adopted three coding sequences for single-layer structures are demonstrated in Figure 3d–f. Moreover, similar coding sequences were utilized to construct double and triple-layer metamaterial structures. Recently, multi-layered structures have become familiar in a few research areas, and they offers unique performance compared to conventional metamaterial. However, the adoption of the multi-layered concept in this analysis demonstrates inconsistent *S*_21_ results, as illustrated in Table 3. For example, Coding Sequences 2 and 3 manifest peak responses between 0 and −20 dB magnitude values for single-layered structure, while the performance reduced and then increased when the number of layers increased. Moreover, for the triple-layered structures, the performances occurred between the ranges from 0 to −35 dB. In summary, the comparison revealed that the performance of the metamaterial is directly dependent on the metamaterial structure, coding sequence, and number of metamaterial layers.

#### 3.2.2. RCS Reduction

A radar cross-section (RCS) is generally used to measure how detectable an object is by the radars and is also known as the electromagnetic signature of the object. In addition, the RCS application is primarily found in military technology because the aircraft wants to penetrate the enemy area without being detected. If the aircraft is detected, then it will be easily destroyed by any anti-aircraft missiles. Hence, military technology has generally utilized this RCS application, and it has also been used in aviation technology. Therefore, it is a very important field of study in radar and antenna engineering. Generally, the definition of RCS is dependent on several factors; for example, if a target is located from the radar at a distance of R, then the RCS of the target can be expressed as stated in Equation (6). Moreover, the RCS is primarily measured in terms of square meters or in the more conventional unit known as dBsm (dBm^2^).
(6)$$\sigma =4\pi R^{2}\frac{P_{r}}{P_{i}};$$
where σ = radar cross-section (m^2^ or dBm^2^); *R* = distance between the target and radar receiver (m); $P_{r}$ = backscattered or reflected power density (Watt/m^2^); and $P_{i}$ = incident power density on target (Watt/m^2^). RCS is also known as the ratio of the backscattered power density ($P_{r})$ to the incident power density ($P_{i})$ on the target. The factors that influence the RCS values are the physical geometry and material, the direction of radar, polarization of the scattered signal, and signal frequency. RCS values can be measured into two types, namely monostatic and bistatic RCS. For the monostatic RCS, the transmitter and receiver antennas are located in the same place, while the bistatic RCS, they are located in different positions.

The RCS values demonstrated in Figure 4a–c clearly illustrate almost similar patterns for all three coding sequences and as well as for the multi-layered structures. Each coding sequence manifested RCS values between the ranges from −45 dBm^2^ to −100 dBm^2^ for every layer. However, Coding Sequence 1 for three different layered structures successfully exhibited less than −50 dBm^2^ RCS values at the frequency range from 4 to 5 THz. Moreover, the proposed Coding Sequence 3 with the single-layered structure was able to reach −50 dBm^2^ at 2 THz frequency. In summary, inconsistent RCS reduction patterns were manifested, and they were strongly influenced by both coding sequences and the number of layers.

### 3.3. Tailored Metamaterial Structure

Another novelty of this work is the introduction of a tailored metamaterial structure constructed uniquely in four different ways. Traces of coding metamaterial concepts are presented in these tailored structures. The conventional array metamaterial is typically constructed with a specific number of rows and columns to gain or optimize the performance. However, the authors introduced a unique structure by arranging the substrate material and metamaterial design inconsistently to check the changes in the performances. The size of Designs 1 to 4 as illustrated in Figure 5a–d subsequently increases in each subfigure and exhibit unique outcomes.

#### 3.3.1. Transmission Coefficient

The *S*_21_ comparison of both coding and tailored metamaterial designs revealed contradictory results. In each tailored structure, more than seven resonance frequencies occurred with acceptable magnitude values, as demonstrated in Table 4. For example, TM1 demonstrated a total of nine resonance frequencies that exhibited more than −15 dB magnitude values. However, the slightly larger tailored structures such as TM3 and TM4 had magnitude values that almost reached −60 dB. This phenomenon strongly depends on the arrangement of substrate material and metamaterial structure designs. Further analysis by utilizing this tailored metamaterial can be very useful for other research fields besides RCS reduction applications. This is because the tailored metamaterial design proposed in this work is generally constructed based on the miniaturization concept. Therefore, it can be applied to any device which satisfies the current advanced technological era. In addition, terahertz metamaterial research work is still in the beginning stages, although many previously published papers have successfully reached their objectives.

#### 3.3.2. RCS Reduction

Although the tailored metamaterial designs manifest excellent *S*_21_ results, the RCS reduction values, however, were able to reach −60 dBm^2^ only, as illustrated in Figure 6. In these contacts, the coding metamaterial designs manifested better reduction values in the terahertz frequency range, which support the theory that the coding metamaterial are able to manipulate electromagnetic wave and realize different functionalities. Furthermore, TM4 has the lowest RCS reduction compared to other designs. Moreover, the RCS reduction values are inversely proportional to the dimension of the tailored structure. In other words, the RCS values were decreased when the size of the structure increased.

### 3.4. Scattering Pattern Comparison

The bistatic far-field scattering patterns of all the introduced metamaterial designs are compared in Table 5 and Table 6. The comparison in Table 5 indicates that all of the coding metamaterial designs exhibit similar scattering patterns and have bistatic RCS values between the range of −45 dBm^2^ and −55 dBm^2^. Moreover, the least RCS values occurred for the double-layered coding metamaterial designs, and the optimized reduction value was successfully obtained in Coding Sequence 1. This revealed that with the increasing number of metamaterial layers in the adopted coding sequences, the bistatic RCS values were reduced. On the other hand, the triple-layer structures had slightly higher RCS values than the single-layer, and therefore the values were dependent on the adopted coding sequences and number of layers. Moreover, the tailored metamaterial designs exhibited unique scattering patterns, as illustrated in Table 6. The designs exhibited RCS reduction values in the range between −69 dBm^2^ and −56 dBm^2^. In addition, this comparison table revealed that the size increment in the tailored design structure led to better RCS reduction values and scattering patterns. Overall, both of the proposed metamaterial design structures manifested unique scattering patterns for each coding sequence or tailored design. Therefore, the RCS reduction values are influenced by the metamaterial design structure, coding sequence, and tailored metamaterial designs. In summary, these analyses revealed that the coding metamaterial is more practicable to be used for the terahertz frequency applications. This is because the relatively new coding-based metamaterial research works are still growing among the scientific community, since it is easy to construct based on the unit cell designs with ‘0’ and ‘1’ elements. Furthermore, based on the literature review, it is indicated that limited studies were carried out by adopting the coding-based metamaterial structure for terahertz frequency applications. As discussed earlier, the coding metamaterial can manipulate EM waves by utilizing various coding sequences when compared to conventional metamaterial design. This is proved by the scattering pattern analysis demonstrated in Table 5 and Table 6. Moreover, the conventional metamaterial design can be varied by only changing the number of array cells that are designed based on the required objectives. The resonance frequencies can be varied based on the array cell structure, but it is difficult to realize the different functionalities and scattering patterns properties. On the other hand, the metamaterial design structures can reduce the weight, enlarge the angular range, or expand the bandwidth. Consequently, it is more practicable to be applied in RCS reduction applications. Hence, in this work, a novel coding metamaterial with various construction designs was analyzed for the terahertz frequency applications.

## 4. Conclusions

In this research work, the authors presented the integration of fast-growing coding metamaterial in terahertz frequencies for the RCS reduction applications. All the coding sequences were constructed by adopting a 1-bit pattern with six lattices. Therefore, the proposed coding metamaterial designs consist of two types of unit cells to mimic “0” and “1” elements with 0 and π phase responses and are arranged in a 6 × 6 array structure. In addition, the tailored metamaterial design structures were also investigated in this work to compare the performances. The comparison of both designs indicates unique behaviors. Furthermore, the bistatic scattering patterns of the coding metamaterial and tailored metamaterial designs were analyzed in this investigation. The RCS comparison values of both coding and tailored metamaterial support the theory that the coding metamaterial is able to manipulate electromagnetic waves and realize different functionalities. This is because the tailored metamaterial is only able to exhibit optimized transmission coefficient results compared to coding metamaterial. The coding metamaterial is generally new to the scientific field, and it can be applied in a wide range of applications such as manipulating antennas radiation beams, reducing scattering features of targets, and manifesting various extraordinary properties of metamaterial. Moreover, coding metamaterial in terahertz frequencies can be used in several fields such as radar, communication, and biomedical imaging. Moreover, RCS reduction generally performs a crucial role in stealth technology for aircraft, missiles, ships, and other military vehicles. With smaller RCS, vehicles can better evade radar detection, whether it be from land-based installations, guided weapons, or other vehicles. Overall, the obtained results from these analyses give a clear vision of integration effects of coding and tailored metamaterial for terahertz frequencies. The coding metamaterial possesses better RCS reduction values; hence, it will suitably be fitted for the terahertz frequency. Further extended investigation of tailored metamaterial can be a valuable asset in this technological era, since it possesses unique scattering parameter behaviors.

## Figures and Tables

**Figure 1 materials-15-02777-f001:**
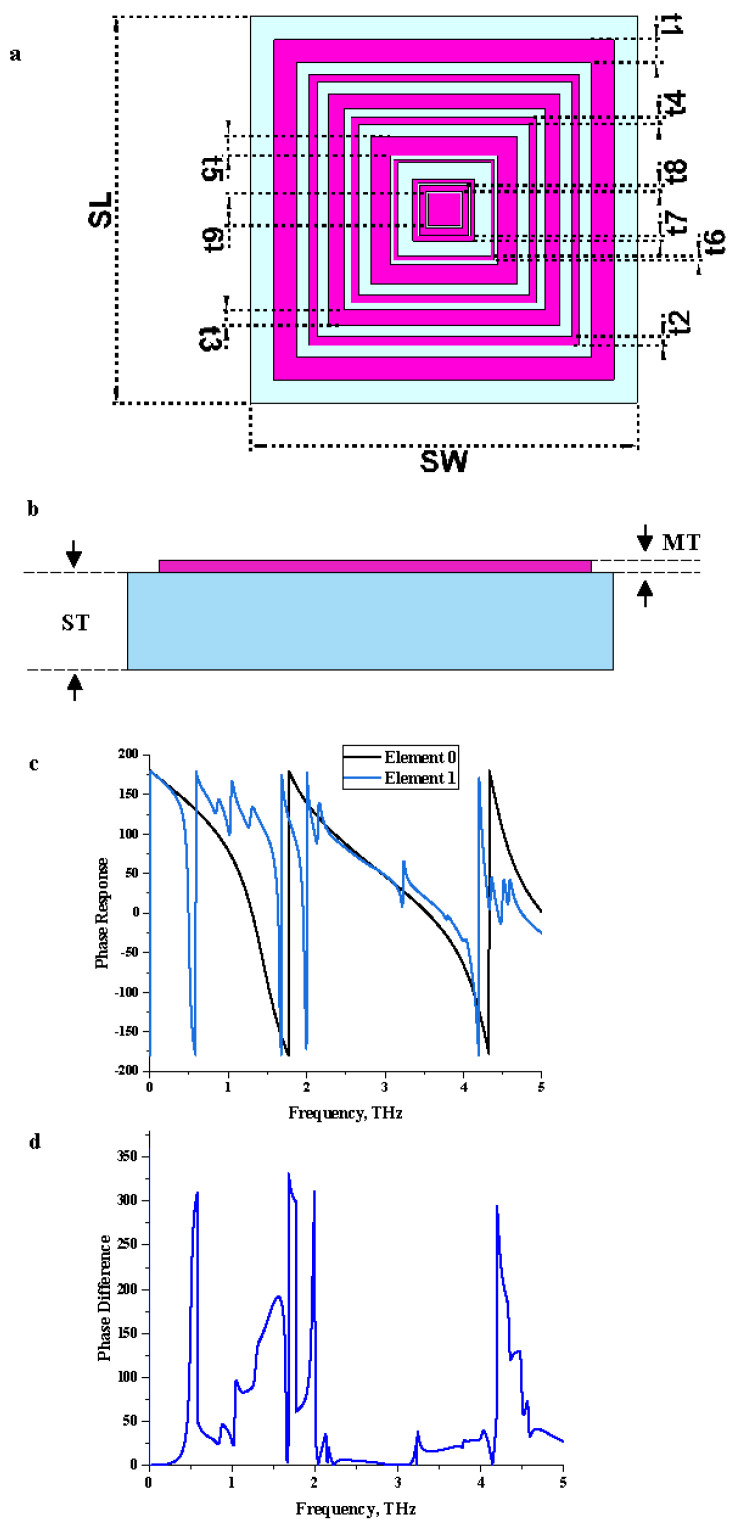
Graphical illustration exported from CST software: (**a**) metamaterial design, (**b**) side view, (**c**) phase response of elements ‘0’ and ‘1’, (**d**) phase difference.

**Figure 2 materials-15-02777-f002:**
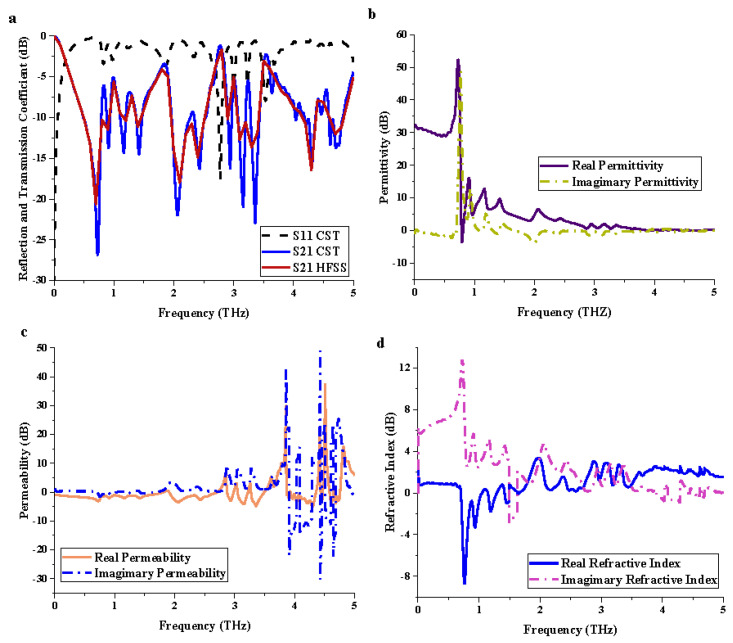
Scattering and effective medium parameters of unit cell metamaterial: (**a**) *S*_11_ and *S*_21_; (**b**) permittivity; (**c**) permeability; (**d**) refractive index.

**Figure 3 materials-15-02777-f003:**
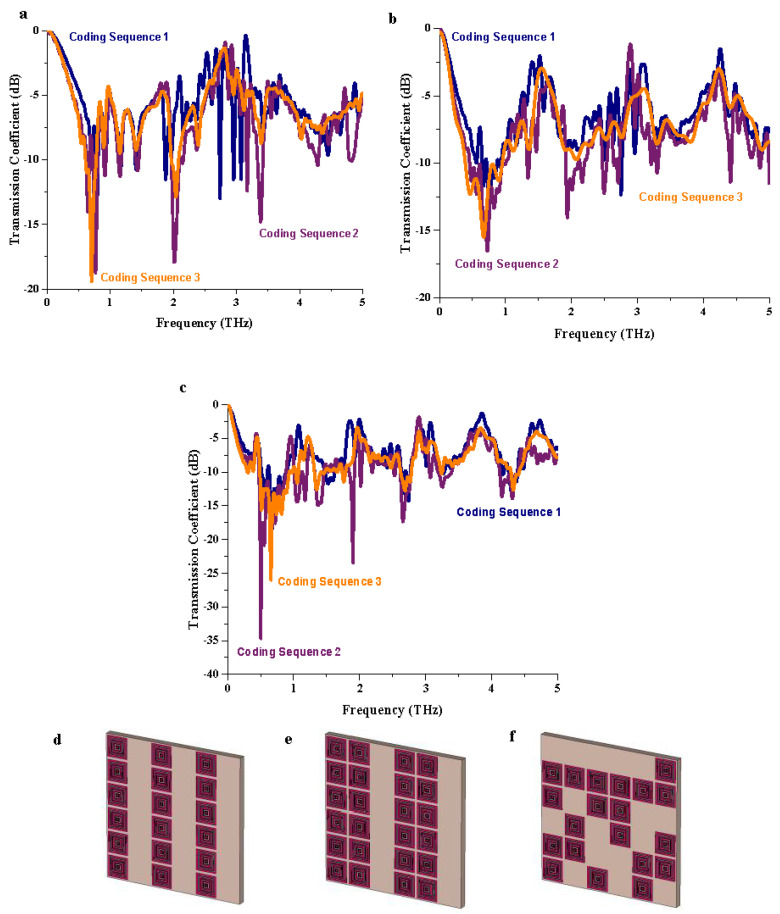
*S*_21_ results of (**a**) single layer, (**b**) double layer, (**c**) triple-layer; single-layer design of (**d**) Coding Sequence 1, (**e**) Coding Sequence 2, (**f**) Coding Sequence 3.

**Figure 4 materials-15-02777-f004:**
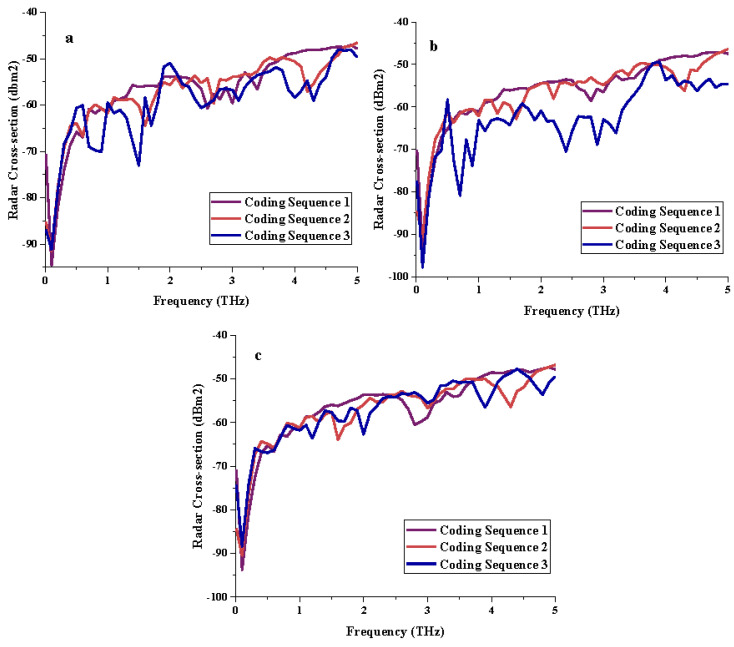
RCS results of three different metamaterial coding sequence for: (**a**) single-layer, (**b**) double-layer, (**c**) triple-layer.

**Figure 5 materials-15-02777-f005:**
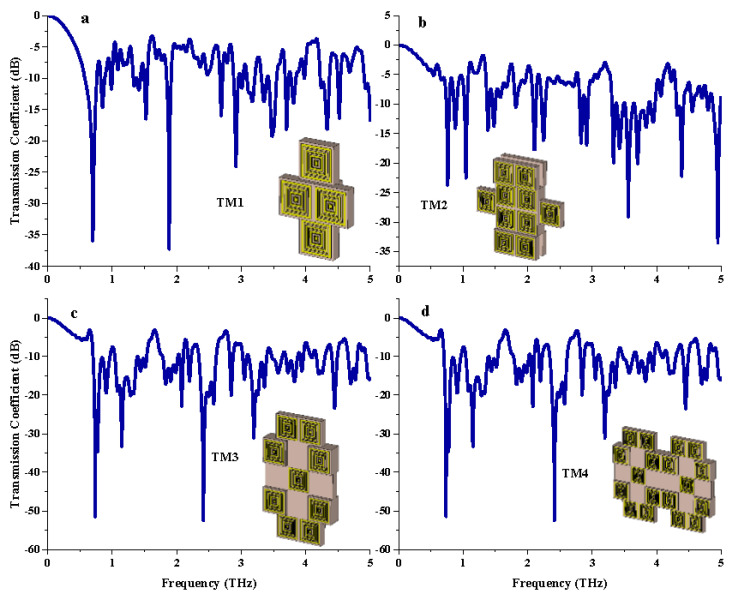
*S*_21_ results of three different tailored metamaterials for: (**a**) TM1, (**b**) TM2, (**c**) TM3, (**d**) TM4.

**Figure 6 materials-15-02777-f006:**
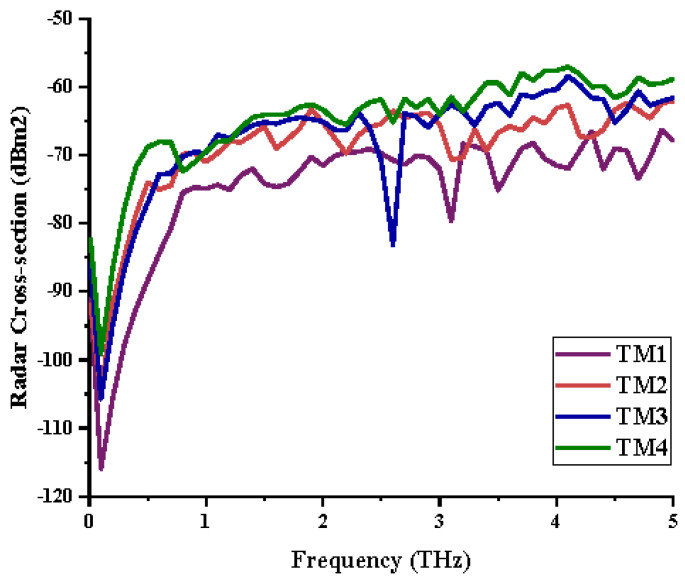
RCS results of four different tailored metamaterials.

**Table 1 materials-15-02777-t001:** Dimension details of the proposed SM design.

Descriptions	Dimension (µm)
t_1_	3.0
t_2_	0.9
t_3_	2.0
t_4_	1.0
t_5_	2.5
t_6_	0.5
t_7_	0.6
t_8_	0.8
t_9_	4.2
Substrate Length, S_L_	50
Substrate Width, S_W_	50
Metamaterial Thickness, M_T_	0.2
Substrate Thickness, S_T_	15

**Table 2 materials-15-02777-t002:** Arrangement of both elements in each proposed coding sequence.

Coding Sequence	Row 1	Row 2	Row 3	Row 4	Row 5	Row 6
1	101010	101010	101010	101010	101010	101010
2	110110	110110	110110	110110	110110	110110
3	101010	110011	010101	101100	111111	000001

**Table 3 materials-15-02777-t003:** Resonance frequencies with magnitude values less than −15 dB for coding metamaterial.

Coding Sequences	Single-Layer	Double-Layer	Multi-Layer
1	-	-	0.67 THz
2	0.77 and 2.02 THz	0.73 THz	0.51, 0.66, 1.90, and 2.66 THz
3	0.70 THz	0.67 THz	0.65 THz

**Table 4 materials-15-02777-t004:** Resonance frequencies with magnitude value less than −15 dB for tailored metamaterial.

Tailored Metamaterial	Resonance Frequencies
TM 1	0.70, 1.53, 1.89, 2.69, 3.48, 3.71, 4.34, and 4.52 THz
TM 2	0.76, 1.05, 2.11, 2.25, 2.83, 2.92, 3.33, 3.43, 3.56, 3.71, and 4.39 THz
TM 3	0.74, 0.91, 1.15, 2.08, 2.42, 2.85, 3.20, and 4.45 THz
TM 4	0.74, 0.78, 1.15, 2.08, 2.42, 3.20, and 4.45 THz

**Table 5 materials-15-02777-t005:** Scattering patterns of coding metamaterial design.

Coding Metamaterial			
	Coding Sequence 1	Coding Sequence 2	Coding Sequence 3
Single-layer	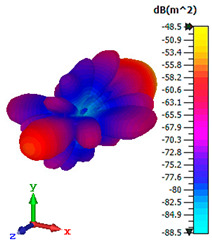	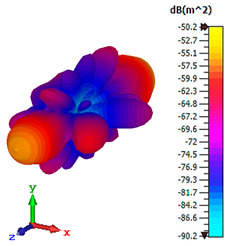	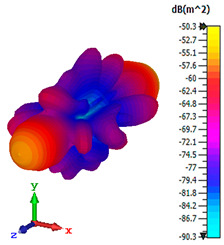
Double-layer	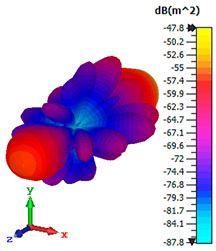	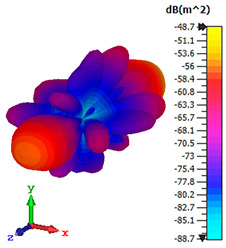	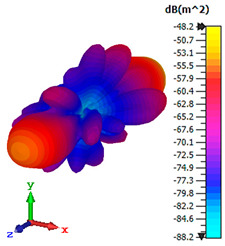
Triple-layer	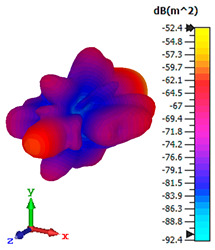	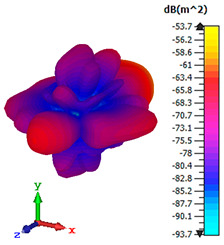	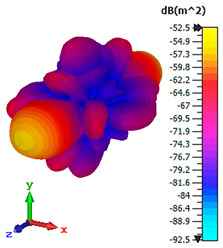

**Table 6 materials-15-02777-t006:** Scattering patterns of tailored metamaterial design.

Tailored Metamaterial			
TM 1	TM 2	TM 3	TM 4
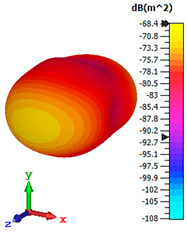	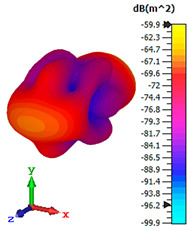	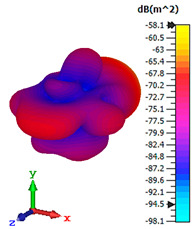	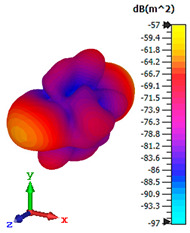

## Data Availability

All the data are available within the manuscript.

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
