# Peer review of "Development and Analysis of Coding and Tailored Metamaterial for Terahertz Frequency Applications"

_materials, 2022, doi:10.3390/ma15082777_

Round 1

Reviewer 1 Report

The manuscript tilted “Development and Analysis of Coding and Tailored Metamaterial for Terahertz Frequency Applications” presents the integration of fast-growing coding metamaterial in terahertz frequencies for the RCS reduction applications. The models were successfully constructed, and they also provide interesting results which can be considered to be applied in radar cross-section (RCS). However, I will wait for revised manuscript before demonstrating my opinion on the publication of this manuscript on the Journal of “Materials”

1.The author should provide some criteria for a metamaterial to be promising for RCS application. Based on these criteria, the authors show that their metamaterials are suitable for RCS application. For example, the coding sequence 1 exhibits  -50 dBm2 RCS value at the frequency range from 4 to 5 THz. Is this an advantageous value for an RCS metamaterial? Or is this value better than other designs in previous studies?

  1. How were the parameters t1, t2, t3, t4, t5, t6, t7, t8, and t9 determined. I meant the values of these parameters, were they chosen based on any criteria?
  2. At the beginning, a CST model does not often produce a resonant frequency. How were t1 – t9 chosen to give a resonant structure?
  3. The meaning of some sentences in the manuscript is not clear, for example:

         4.1. Many research 50

               investigations by adopting metamaterial designs were performed                   past few decades such 51

(Research and investigation are the same thing, we can use just one of them; Research ON adopting metamaterials is correct)

          4.2. Besides that, studies of the metamaterial for terahertz frequencies           also becoming 78

         famous among researchers. In 2010, Iwaszczuk et al. [10] investigate               angle- and frequency- 79

(…is also becoming…or what?)

Therefore, the authors should revise the whole manuscript for any typo, for improving sentences whose meaning are not clear.

Author Response

As attached.

Reviewer 2 Report

The authors study 2 metamaterial structures for THz frequency band. Extensive simulations are presented of different coding metamaterials using CST software.

Positive remarks about the paper:

  • The paper’s subject is relevant to the journal
  • Comparisons of the results from 2 different simulation platforms (HFSS, CST) are presented
  • It is well written in clear, idiomatic English.
  • Previous related work is adequately referenced.
  • It contains a lot of results, which might be really useful for researchers in the field.
  • The keywords accurately reflect the content.

Minor issues

  • The results are not accompanied by experimental measurements.

My overall impression is that the manuscript is suitable for publication as authors taking into consideration the above minor issue.

Author Response

As attached.

Reviewer 3 Report

See attached review

Author Response

As attached.

Reviewer 4 Report

This paper presented comparative research on coding and tailored metamaterials for optimal RCS reduction in the terahertz range. The theoretical analysis is poor, and the simulation results can not support its conclusion. The lethal failure is that the RCS reduction of all the designs does not reflect the diffusing principle of destructive interference on the coding array. I decline the manuscript for publication based on the following considerations.

  1. What is the difference of the reflection coefficient with both amplitude and phase between “0” and “1” elements? It is unbelievable to see huge fluctuations in the reflection and transmission coefficients.
  2. What is the purpose of calculating effective permittivity and permeability for RCS reduction?
  3. It is unpractical and meaningless to use miniaturized and irregular arrays.
  4. The conclusion is unclear for the different coding sequences and array sizes show slight differences.

Author Response

As attached.

Round 2

Reviewer 2 Report

The authors explained sufficiently  why measurements are not included. My overall impression is that the paper can be published.

Reviewer 3 Report

The manuscript has undergone the required corrections and can be accepted.